# A Survey on the Expression of the Ubiquitin Proteasome System Components HECT- and RBR-E3 Ubiquitin Ligases and E2 Ubiquitin-Conjugating and E1 Ubiquitin-Activating Enzymes during Human Brain Development

**DOI:** 10.3390/ijms25042361

**Published:** 2024-02-17

**Authors:** Stefano Magnati, Eugenio Alladio, Enrico Bracco

**Affiliations:** 1Centro Regionale Anti Doping—A. Bertinaria, Orbassano, 10043 Turin, Italy; eugenio.alladio@unito.it; 2Politecnico di Torino, 10129, Turin, Italy; 3Department of Chemistry, University of Turin, 10125 Turin, Italy; 4Department of Oncology, University of Turin, 10043 Orbassano, Italy; 5Istituto Nazionale Ricerca Metrologica, 10135 Turin, Italy

**Keywords:** brain development, neurodevelopmental disorders, neurodegenerative disorders, ubiquitin proteasome system (UPS), proteome remodeling, machine learning and model classification

## Abstract

Human brain development involves a tightly regulated sequence of events that starts shortly after conception and continues up to adolescence. Before birth, neurogenesis occurs, implying an extensive differentiation process, sustained by changes in the gene expression profile alongside proteome remodeling, regulated by the ubiquitin proteasome system (UPS) and autophagy. The latter processes rely on the selective tagging with ubiquitin of the proteins that must be disposed of. E3 ubiquitin ligases accomplish the selective recognition of the target proteins. At the late stage of neurogenesis, the brain starts to take shape, and neurons migrate to their designated locations. After birth, neuronal myelination occurs, and, in parallel, neurons form connections among each other throughout the synaptogenesis process. Due to the malfunctioning of UPS components, aberrant brain development at the very early stages leads to neurodevelopmental disorders. Through deep data mining and analysis and by taking advantage of machine learning-based models, we mapped the transcriptomic profile of the genes encoding HECT- and ring-between-ring (RBR)-E3 ubiquitin ligases as well as E2 ubiquitin-conjugating and E1 ubiquitin-activating enzymes during human brain development, from early post-conception to adulthood. The inquiry outcomes unveiled some implications for neurodevelopment-related disorders.

## 1. Introduction

Brain development begins shortly after conception, with critical processes occurring during prenatal and early postnatal stages. The human brain is a complex organ composed of billions of neurons and glial cells, and its development involves intricate and highly regulated sequences of events [1]. Neurogenesis occurs during prenatal development, which is the process of generating mature neurons (Figure 1).

The differentiation process foresees several changes including proteome remodeling, which is the result of the neosynthesis of proteins because of transcriptional changes [2,3,4] and of protein disposal, handled by the evolutionarily conserved ubiquitin proteasome system (UPS) [5,6] upon selective protein ubiquitination. As the brain takes shape, neuronal migration is a critical process. Cells move to their designated locations, accurately shaping the brain’s structure. Hence, cell migration is a vital component of brain development as it establishes precise patterns of connections between nerve cells and is crucial for axonal guidance and neurite outgrowth [7]. From birth through early and middle childhood, neuronal myelination takes place. Myelination involves the formation of a fatty insulation called myelin around the axons of neurons, which increases the speed and efficiency of neural communication [8,9,10]. In parallel with myelination, neurons form connections with one another through synapses [11,12]. Over time, these connections are refined and pruned, a process that fine-tunes the neural circuits and optimizes their functioning. Synaptic pruning continues throughout adolescence [13].

A finely tuned program genetically controls the different processes driving brain development [4]. The genes orchestrating these processes play a crucial role in ensuring that the brain forms correctly and functions optimally.

Perturbations occurring at any of the early stages of brain development can lead to the emergence of neurodevelopmental disorders (NDDs) [14]. NDDs encompass a wide range of conditions, each with unique features [15]. These disorders are often associated with genetic or environmental factors that alter the typical brain development [14].

The selective protein disposal executed by the UPS relies on a coordinated and stepwise action executed by three classes of enzymes to label a protein destined for degradation with a polyubiquitin (poly-Ub) chain [5,6]. The enzymatic cascade leading to flagging a protein with poly-Ub starts by activating ubiquitin. This first step is energy-dependent, relying on ATP hydrolysis, and is catalyzed by E1 Ub-activating enzymes, which link the Ub C-terminus to a cysteine residue of the enzyme. Afterwards, the activated Ub is transferred as a thioester from E1 to a cysteine residue present on the catalytic site of E2-conjugating enzymes. Ultimately, E3 Ub ligases catalyze the very last step by recruiting both E2 and the substrate and promoting the covalent attachment of Ub to the substrate. Usually, Ub is conjugated to a Lys residue on the target protein, though less frequently can be attached to other amino acid residues (i.e., N-terminal Met, Ser/Thr, and Cys) [16,17]. The processivity of the ubiquitination reaction ensures the formation of a poly-Ub chain. In conjunction with E2, E3 Ub ligase family members provide substrate specificity. Based on their catalytic and 3D structural features, the E3 family members are catalogued into three distinct families: RING (really interesting new gene), HECT (homologous to the E6AP carboxyl terminus), and RBR (ring between ring) [18]. Unlike RING, HECT and RBR family members catalyze Ub transfer directly to the substrate through a two-step reaction. Ub is first transferred to the catalytic cysteine residue on E3 and then covalently attached to the substrate. HECT-E3 can function alone or in conjunction with accessory or adapter proteins. The substrate selectivity and specificity of the RING members require the assembly of multi-subunit complexes, whereas, for the HECT and RBR members, the N-terminal region is sufficient for substrate recognition. Remarkably, an individual E3 can recruit more than one substrate, while on the contrary, any substrate may be targeted by more than one E3 ligase [19,20]. Due to their intrinsic catalytic activity toward the substrate/s, the E3 HECT family members are, among all the E3 members, the most suitable druggable candidates [21,22,23]. Ubiquitination is a very versatile post-translational modification. Indeed, Ub can be conjugated directly to a target protein or to itself through either one of its seven conserved Lys residues (Lys6, Lys11, Lys27, Lys29, Lys33, Lys48, and Lys63) or the N-terminal Met1 residue, thus leading to structurally distinct types of poly-Ub chains, linear or branched, of different length, giving rise to an extremely broad Ub lexicon [24,25,26]. Usually, proteins modified with poly-Ub chains internally linked through K11 or K48 appear to be the favored proteasome substrates [27,28]. The 26S proteasome is a 2.5 MDa complex composed of different protein subunits arranged into an elongated structure composed of a central 20S core particle (CP) with one or two terminal 19S regulatory particle(s) (RP(s)) [29,30]. Ubiquitinated proteins are recognized by the RP and, through an ATP-dependent mechanism, are unfolded and then channeled to the catalytic CP, where the inner beta subunits, thanks to their proteolytic activities, ultimately degrade them.

Over the past years, several components of the UPS have been reported to play pivotal roles in various aspects of neuronal development, including dendrite, axon, and synapse morphogenesis [31,32]. Additionally, some forms of NDDs, including syndromic forms of autism spectrum disorder such as Angelman syndrome and facial dysmorphism, macrocephaly, and intellectual disability are associated with mutations in HECT-E3 Ub ligase-encoding genes [33,34,35,36,37,38,39,40]. Ultimately, continuous stresses and ageing lead to an intracellular accumulation of proteinaceous aggregates (altered proteostasis). To avoid harmful changes, the UPS and autophagy processes facilitate the degradation of unwanted misfolded proteins that would otherwise contribute to the formation of aggregates. Consistently, impairment in the UPS is often associated with neurodegenerative disorders, such as Parkinson’s disease [41,42].

Through deep data mining and analysis and by taking advantage of machine learning-based models, we first mapped the transcriptomic profiles of the genes encoding HECT- and RBR-E3 ubiquitin ligases as well as E2 ubiquitin-conjugating and E1 ubiquitin-activating enzymes during human brain development, from early post-conception to adulthood. We then explored the transcriptional changes occurring throughout brain development, employing high-dimensional data analysis. Eventually, and for the first time, we validated the data by machine learning-based tools. The outcomes from this inquiry will contribute to increasing our understanding of the role of the UPS underlying human brain development and will unveil some implications for neurodevelopment-related disorders.

## 2. Results

### 2.1. Gene Expression Profiling of the E1, E2, and E3 HECT and RBR Members Displays Significant Spatio-Temporal Changes

Brain development, including that of human beings, is very complex, involving several processes accomplished in genetically predetermined and sequential steps. Among them, are neuron precursor differentiation, migration, and myelination and synaptogenesis, where the UPS members are fundamental players involved in proteome remodeling [43,44]. Accordingly, their malfunctioning has been associated with neurological disorders, including neurodevelopmental and neurodegenerative disorders, with several degrees of severity [45]. By taking advantage of publicly available databases, we surveyed the transcriptional signature of several UPS members, including E1, E2, and some members of the E3 family (i.e., HECT and RBR) during human brain development. The whole list of the genes surveyed, alongside those displaying the most noticeable changes, is summarized in Appendix A. The time window spanned the whole life cycle, from the very early prenatal stages (8 post-conceptional weeks, pcw) to adulthood. When analyzing the developmental transcriptome data, profiled according to age, we noticed that most of the genes were expressed, though to different extents, throughout the whole timeframe analyzed (Figure 2A). Remarkably, the mRNAs amount differed significantly over the timeline.

While some genes exhibited elevated expression levels (e.g., *UBA1*, *UBE2M*, *UBE2D2*, *CDC34*), others displayed extremely low, if any, expression levels (e.g., *UBE2U*) throughout the entire life cycle. Surprisingly, all the E1, E2, and E3 members we analyzed exhibited statistically significant differential gene expression at at least one of the time points. Hierarchical cluster analysis unambiguously indicated that sharp changes occurred at birth, with a boundary line separating the prenatal, except for the late prenatal group, and postnatal stages. Additionally, adulthood and adolescence clustered together at a closer analysis, similarly to infancy and early childhood. An appreciable decline, though to different extents, in mRNAs amount from prenatal to postnatal stages was observed for several genes including *UBE2T*, *UBE2C*, *Nedd4L*, *UBE3A*, *RNF19B*, *NAE1*, *UBE2Z*, *UBE2F*, *UBE2D1*, *UBE2L6*, *UBE2R*2, *UBE2S*, *BIRC6*, *HECW1*, *UBR5*, and *UBA6*. Conversely, we noticed that a smaller group of genes displayed the opposite pattern, being upregulated during the postnatal stages when compared to prenatal ones. We detected *AKTIP*, *UBE2E2*, *Herc6*, *RNF144B*, *UBA7*, *CUL9*, and *PARK2* in the latter group.

We then examined mRNA expression in the different brain regions and compared the determined expression levels. The differences observed were less subtle compared to those found in the temporal analysis. Indeed, the number of genes differentially expressed dropped sharply (Figure 2B).

Interestingly, *Nedd4*, *UBE2I*, *UBA2*, and *SMURF2* exhibited a pattern opposite to those of *UBE2E2* and *RNF19B*. Remarkably, hierarchical clustering revealed that ganglionic eminence areas (i.e., CGE, LGE, and MGE) clustered with neocortical regions (i.e., TCX, PCX, and OCX), being the former transient structures from which during fetal brain development, several neuron types are differentiated and that contribute to originate the neocortex. Worthy of note, the primary visual and auditory cortex areas (i.e., A1C and V1C) clustered very close.

Overall, the mRNA amount of the genes encoding the E1, E2, and HECT- and RBR-E3 family members were spatio-temporally regulated. While the differences between the prenatal and the postnatal stages were quite pronounced, those among the different regions appeared more subtle, though appreciable.

### 2.2. Correlation Analysis Defines Coregulated E1, E2, and E3 in the Brain

Since the UPS relies on a concerted action of multiple players lying into a cascade of reactions, in which the activity of E1 activating enzymes precedes that of E2 conjugating enzymes, which is eventually followed by the intervention of E3 ubiquitin ligases, we attempted to determine any potential correlation between E1, E2, and E3. The inquiry aimed to identify a putative coregulation of the different E1, E2, and E3 members, which might provide insights to outline which E1 might support the activity of E2 and which E3 might be supported by which E2. The correlation assessments, aimed to unveil a potential biologically relevant gene coregulation, were performed using the variance inflation factor (VIF) and the Spearman test and applying a stringent threshold arbitrarily set at 0.7. Prior to this, the data were preliminarily surveyed by an exploratory analysis aimed at assessing the fundamental assumptions of normality, homoscedasticity, and correlation. The outcome highlighted that the data neither followed a normal distribution nor exhibited homoscedasticity. Then, the analysis was initially carried out on the whole dataset by averaging the different gene expression values from the different age groups (i.e., early prenatal stage, infancy, early childhood, adolescence, adulthood, etc.), thus enabling the exploration of a potential correlation independently of the variable “age group”. Interestingly, we observed that only a single E1, *ATG7*, did not correlate, whereas a single member, *UBA7*, anti-correlated with most of the E2-encoding genes, and many positive correlations were identified (e.g., *UBA3* vs. *UBE2N*/*UBE2D1*/*UBE2Q2*) (Figure 3A). When the correlation analysis was performed between E2 and E3 members, the positive correlation events increased sharply (Figure 3B), also because the magnitude of the matrix increased considerably.

Overall, the analysis revealed that quite a few E1, E2, and E3 members displayed coregulated gene expression, thus letting surmise a few putative brain-specific E1, E2, and E3 cascades, including (E1)UBA6 → (E2)UBE2R2 → G2E3/ARIH1/UBR5/SMURF2/Nedd4L/UBE3A(E3); (E1)UBA6 → (E2)UBE2J1 → (E3)G2E3; (E1)UBA1 → (E2)UBE2I → (E3)Nedd4L/G2E3; (E1)UBA6 → (E2)BIRC6 → (E3)Herc1; and (E1)UBA5 → (E2)UBE2N/UBE2K → (E3)UBE3A.

We then explored whether, if any, the identified correlation patterns could be affected by age. Remarkably, the survey outcome indicated robust correlations among different age groups, particularly between the early childhood and the adolescence groups (Appendix A), while when the other age groups were analyzed no significant correlations (Appendix A) were noticeable.

### 2.3. Clustering Reveals That the Brain Gene Expression Signatures of E1, -E2, and RBR- and HECT-E3 Differ Significantly between Prenatal and Postnatal Stages

Based on the cluster map analysis, we first attempted to dissect the data through unsupervised cluster analysis to determine whether specific gene expression profiles could feature different ages.

For this purpose, to assess the presence of potential clusters as envisaged by the cluster maps, we conducted gene expression data dimensionality reduction using t-distributed stochastic neighbor embedding (t-SNE). When the data were projected in two dimensions (2Ds), at least two significant clusters were detected (Figure 4A), one of which was more sharply defined when compared to the other.

Afterwards, to assess whether the “age group” variable influenced the clustering, the different clusters were color-coded according to the different age groups. A sharp separation between prenatal and postnatal groups was observed, with the only exception of the late prenatal group. Indeed, the latter appeared closer to the postnatal cluster than to the prenatal one, suggesting that, at least in terms of some UPS components (i.e., E1, E2, and RBR- and HECT-E3), a sort of “steady-state” gene expression signature was reached already shortly before birth. Notably, the prenatal cluster exhibited a higher degree of compactness and homogeneity when compared to its postnatal counterpart, which was even more appreciable when the data were projected in three dimensions (3Ds) (https://github.com/SMagnati/SMagnati, accessed on 1 March 2023). Noteworthy, we could not rule out the presence of other small and less defined sub-clusters in the postnatal cluster. Likely, some sub-clusters were still present in the postnatal group, though they were less evident when compared to the markedly different clusters observed before and after birth. We then further mapped the same cluster restricting the analysis to two categorical variables, namely, ethnicity and gender (Figure 4B,C).

The outcome revealed that both variables strictly influenced the intra-clustering ability. To verify this hypothesis, we performed t-SNE on two subsets of the original dataset, filtering the data for the most representative ethnicity (European) and a single gender (i.e., male), respectively. Notably, gender intra-clustering, when compared to ethnicity intra-clustering, displayed a markedly lower variability in the postnatal cluster.

Hence, when surveying E1, E2, and HECT- and RBR-E3 members’ gene expression, birth represented a key turning point. We then intended to appraise the significantly differentially expressed genes between the prenatal (i.e., early, early-mid, late-mid prenatal stages) and the postnatal (i.e., adolescence, adulthood, childhood, and infancy) groups.

Since the “late prenatal” group clustered distantly from its related prenatal groups and because of the small size of the group (*n* = 22), we opted to exclude the “late prenatal” group from the analysis. Leveraging a log_2_-transformed data structure, we calculated the fold change (FC) by simply taking the difference between the postnatal and the prenatal groups. Finally, we summarized the data with a volcano plot (Figure 5A), where the positive values indicated the genes whose expression was upregulated in the postnatal group.

Notably, the mRNA amount of ten genes was significantly decreased proceeding from prenatal age to the postnatal stage (i.e., *UBE2C*, *UBE2R2*, *G2E3*, *UBR5*, *SMURF2*, *UBE2S*, etc.). Conversely, only three genes exhibited higher expression levels (i.e., *PARK*, *AKTIP*, and *Herc6*) in the prenatal group compared to the postnatal group. Interestingly, among the downregulated genes, most encoded E2 conjugating enzymes (6 out of 10). On the contrary, all the upregulated genes encoded primarily E3 members. Noteworthy, these findings are in line with the previous correlation inquiry.

When the same survey was performed exclusively on the postnatal cluster, by dividing it into two classes, i.e., (a) a “children” group, including “infancy”, “early childhood”, and “late childhood”; and (b) an “adults” group, encompassing “adolescence” and “adulthood”, the volcano plot (Figure 5B) did not reveal any significant difference, aligning with previous observations in the cluster analysis and strengthening the previous suggestion that, at least in terms of gene expression of some UPS components (i.e., E1, E2, and HECT- and RBR-E3), birth represents a pivotal turning point for brain development.

### 2.4. Data Validation by Machine Learning-Based Model Classification

Due to the paucity of the material, alongside the technical difficulties in collecting healthy fresh specimens from the different human cerebral areas at different ages encompassing the whole life cycle from the post-conception fetal stage to adulthood, experimental validation was infeasible. Hence, we attempted to validate and strengthen our findings by using machine learning approaches and eventually assessing the robustness of the classification models. Among others, when compared to and differently from a clustering approach that relies on unsupervised learning, this approach offered the benefit of combining data validation with the possibility to identify putative “hidden neighbors” within the whole dataset.

The outcomes of each model’s performance metrics concerning the training and testing sets are summarized in Table 1. The table offers a concise overview of key evaluation metrics for the various classification models, where accuracy reflects the overall correctness, precision estimates the accuracy of the positive predictions, recall measures the ability to capture all positive instances, and the F1-score strikes a balance between precision and recall. A comprehensive analysis was performed to assess the models’ effectiveness on both training and testing datasets.

Remarkably the outcome of the machine learning inquiry highlighted that minimal errors were primarily restricted within classes 0 and 1, representing the postnatal groups. This finding was further supported by the area under the curve (AUC) plot (Figure 6A), which depicts the receiver operating characteristic (ROC) curve, by plotting the true positive rate (sensitivity) against the false positive rate (1-Specificity) at various threshold settings.

The curve’s proximity to the upper-left corner indicated the model’s robust ability to discriminate between different age groups, with XGBoost displaying commendable performance. To verify and further strengthen our assessment beyond the confines of confusion matrix metrics, we evaluated the model’s performance by introducing the error threshold of XGBoost after 10 iterations (Figure 6B).

Notably, it appeared that during the training session, the model exhibited substantial “learning activity” up to the 20th iteration, after which a plateau was observed for both training and testing sessions. For machine learning-based algorithms to achieve widespread acceptance, the identification of the features driving the prediction is also crucial. Hence, the analysis was further improved by assessing the importance of the features influencing the model’s decision for each class by taking advantage of the SHAP library. Using SHAP values, the most discriminatory genes were identified. Interestingly many of the latter were also identified by the previous differential spatio-temporal gene expression analysis. Among them, *Nedd4L* and *UBE2I* appeared as excellent candidates in discriminating the adult/adolescent (class 0) and prenatal (class 2) groups, being their mRNA amounts lowered with increasing age (Figure 6C).

This insightful analysis shed light on the critical factors guiding the model’s predictions across different classes, offering a more comprehensive understanding of its decision-making process.

## 3. Discussion

Human brain development starts shortly after conception and becomes mature in adolescence [1]. The whole process is under tight genetic control [4]. Once brain development ends, most of the neurons remain as post-mitotic cells, thus relying on a finely tuned and coordinated maintenance program in which the UPS and autophagic flux dispose of damaged and obsolete proteins and subcellular compartments [45,46,47,48,49]. Additionally, during adulthood and adolescence, the UPS is actively involved in neuron-specific processes, including synaptic plasticity and homeostatic scaling [43,50,51]. Regrettably, with ageing, the machinery in charge of the disposal of damaged proteins and organelles (i.e., the UPS and autophagy) reduces its activity, which thus increases the risk of accumulating cytosolic proteinaceous and insoluble toxic aggregates that in turn cause cell injury and death. Consistently, those aggregates represent a landmark of neurodegenerative disorders, including Parkinson’s and Alzheimer’s disorders [52,53,54].

The outcome of the survey herein presented revealed that genes encoding for the UPS members E1, E2, and HECT- and RBR-E3 were spatio-temporally and differentially expressed during brain development. While the expression of some members was barely detectable (i.e., *UBE2U*), that of others (i.e., *UBA1*) displayed quite high levels throughout the whole timeframe (i.e., from 8 pcw to adulthood). Interestingly, we detected marked temporal differences, while the differences ascertained among the various brain regions were more subtle, though statistically significant. Overall, but not surprisingly, the E1 members did not display considerable changes over time and, to different extents, appeared constitutively expressed. Plausibly, this is consistent with the role played by the E1 members in the ubiquitination process. Indeed, the E1 members catalyze the first step, and each of them might serve several downstream E2 members, which in turn might amplify the signal by activating an even larger number of E3 enzymes (more than 600) [55]. Therefore, E1 constitutive expression is a prerequisite for letting the downstream effectors, namely, E2 and then E3, properly execute their action. Accordingly, the loss of function of most of the E1 members, including *UBA1*, *UBA6*, and *UBA5*, leads to neuronal dysfunction at various levels and, to some extent, phenocopies the E3 loss of function [56,57,58]. Interestingly, while for some HECT-E3 members, including *UBE3A*, *HACE1*, *ITCH*, *Nedd4L*, and *HACE1*, a robust association between gene mutation and NDDs has been established, our survey indicated that their gene expression underwent dramatic changes at birth. Currently, the mechanistic meaning of this event remains unknown, and further studies are required to ascertain it. The issue will probably be clarified when we gain some more broad insights into the prenatal and postnatal timing and selectivity of the E3 substrate/s. Furthermore, among the different genes that were differentially expressed, we also identified the large Herc family members *Herc1* and *Herc2*. Though their temporal changes in terms of gene expression were milder, when compared to those of *UBE3A* or *Nedd4L*, large Herc members have recently gained relevance because when their encoding genes are mutated, they are associated with clinical syndromes closely related to NDDs, resulting in intellectual disability, dementia, epileptic seizures, and/or signs of autism [59]. Murine models have helped to clarify the mechanisms underlying the phenotypes and progressive degeneration of Purkinje cells; defects at the neuromuscular junction and impaired motor control have been described [60]. The role of the large Herc members in controlling cell differentiation is supported by broad observations in tissues other than the nervous tissue. For instance, it was recently reported that large Herc members might be involved in osteoclast and myeloid cell differentiation [61,62]. Whether large Herc family members are involved in the removal of crucial pro-stemness proteins, thus favoring neuronal differentiation, could be a challenging issue to be ascertained.

The outcome from the clustering analysis provided some interesting hints. Indeed, besides the sharp separation between prenatal and postnatal age groups, it appeared that infancy and early childhood clustered close to each other. This is consistent with the environmental sensory experiences of human beings at that age [63,64,65] and currently lead to hypothesizing a potential role of some components of the UPS and the autophagy flux in adapting brain functions and synapses to those stimuli. Since, on this side, our understanding is still limited, we suppose that our findings might represent a good starting point to unravel the issue.

The correlation analysis laid the basis to suppose potential brain-specific E1 → E2 → E3 axes. While most of the E1 and E3 members have been associated with different NDDs, less is known about E2 members. Our findings might represent an initial steppingstone to experimentally validate the identified potential neuron-specific E1 → E2 → E3 axes.

Because of the scarcity of the material, due to the technical difficulties in collecting healthy fresh specimens from the different human cerebral areas at different ages ranging from the early post-conception fetal stage to adulthood, we attempted to validate and strengthen our findings by using, for the first time, machine learning classifier approaches and eventually by assessing the importance of the genes that contributed to the prediction of the classes by computing Shapley values. Shapley values were used as explanatory variables for the machine learning-based classifiers to corroborate the association between feature importance and predictive performance. This model provided results that were rather consistent with those obtained with the differential spatio-temporal gene expression analysis, thus further strengthening the classification performance. To our knowledge, this is the first report in which this approach was applied to the brain developmental transcriptome.

Overall, coupling machine learning classifiers approaches and feature importance analysis would be of great benefit for those settings in which paucity of the targeted material represents a limitation.

## 4. Materials and Methods

### 4.1. Dataset Collection

We employed a publicly available dataset from the open access repository Brain Span (https://www.brainspan.org/, accessed on 1 March 2023), which provides comprehensive gene expression datasets generated using the mRNA sequencing technology.

The analytical pipeline used to carry out the inquiry is detailed in Figure 2 and, basically, involved the combinational employment of statistical analyses and machine learning approaches.

The Brain Span dataset comprises 504 observations across 89 genes, i.e., 9 E1, 37 E2, 14 RBR-E3, and 29 HECT-E3. To comply with Brain Span’s predefined categorization, we grouped ages into nine representative categories: early prenatal, early mid-prenatal, late-mid prenatal, late prenatal, infancy, early childhood, late childhood, adolescence, and adulthood.

Beyond age-related details, this dataset includes additional variables, such as brain regions, gender, and ethnicity of the donors, providing a robust foundation for analyses.

All analyses and implementations were conducted using Python (www.python.org, accessed on 1 March 2023). The entire codebase and comprehensive workflow have been made openly accessible on GitHub (https://github.com/SMagnati/SMagnati, accessed on 1 March 2023) to facilitate transparency and enable others to scrutinize the database with different queries.

### 4.2. Exploratory Data Analysis

The dataset underwent minimal preprocessing, as the original data already exhibited high accuracy. The gene expression data were transformed into log_2_ format, and upon examination, neither NaN (not a number) values nor notable outliers were detected. A few missing values were present in the donor ethnicity and donor gender variables that, for consistency, were subsequently labelled as “Unknown”.

To gain insights into the characteristics of the data, we conducted a thorough analysis to assess the foundational assumptions of normality, homoscedasticity, and correlation. This step aimed to ensure the reliability and suitability of the dataset for subsequent analyses.

### 4.3. Data Analysis

A series of multivariate statistical tests were conducted to determine potential significant differences in gene expression across various categorical variables, including age, ethnicity, gender, and brain regions. The selection between parametric and non-parametric tests was guided by insights gleaned from an exploratory data analysis. For further details, refer to the pipeline depicted in Figure 7.

For non-parametric tests, the Kruskal–Wallis’s test was employed. A thorough post hoc analysis was executed using the Games–Howell procedure after obtaining statistically significant results. On the contrary, parametric tests, specifically, the ANOVA and, when applicable, the *t*-test, were employed to probe distinctions among groups. This comprehensive methodology ensured a robust exploration of gene expression patterns, considering the diverse nature of the categorical variables.

To mitigate the risk of type 1 errors (i.e., detecting significant differences solely due to data dimensionality), we adjusted the *p*-values using the Benjamin–Hochberg procedure. For a more stringent statistical analysis, we developed and deposited, into GitHub, a function that also considered the confidence intervals.

Clustering analyses were implemented to uncover potential groupings based on the dataset variables visually. To determine the presence of potentially up- or downregulated genes within the clusters, a data visualization technique, known as a “volcano plot”, was employed. Data are presented as means for each gene within both classes analyzed and statistically significant differences, assessed by calculating the *p*-values using the Mann–Whitney U test. False positives were controlled by applying false discovery rate (FDR) adjustments through the Benjamin–Hochberg test. Positive values indicated the genes whose expression was upregulated in the postnatal group.

### 4.4. Data Validation

For data validation, a new target variable was designated with the following classes:Class 0: encompassing “adolescence” and “adulthood”;Class 1: “infancy”, “early childhood”, and “late childhood”;Class 2: the remaining prenatal groups (i.e., early prenatal, early-mid prenatal, late-mid prenatal).

The “late prenatal” group was excluded due to its (a) borderline behavior and (b) small sample size (*n* = 22), when compared to the other age groups.

Since the categorical variables did not exhibit homogeneity across the three target classes, only the numerical variables related to gene expression were considered to avoid artificial overfitting of the model. This approach ensured a more robust analysis, preventing the model from capturing spurious correlations and enhancing its generalization capabilities.

An initial preprocessing step involved the application of min-max normalization to the whole set of features, ensuring standardization to a uniform range. Subsequently, the dataset was partitioned into training and testing sets, employing an 80–20 split, preserving the class proportions within each subset. Initially, as supervised algorithms for the model classification, the k-neighbor classification (KNN), random forest (RF), and support vector classifier (SVC) were employed. Afterwards, the hyperparameters were finely tuned to optimize model performance through 5-fold cross-validation. Continuous training and test performance assessment was conducted to identify and address any signs of potential overfitting. Thereafter, we implemented advanced machine learning techniques such as the bagging classifier and ensemble voting classifier. To refine our modelling approach, boosting was incorporated, utilizing XGBoost. This comprehensive strategy ensured a robust evaluation and enhancement of model performance, exploiting a combination of established and advanced methodologies. The evaluation of model performance was conducted using metrics derived from confusion matrices. This assessment aimed to determine the classification accuracy and effectiveness of the models for both the training and the testing datasets. The confusion matrix provided a detailed breakdown of the true positive (TP), true negative (TN), false positive (FP), and false negative (FN) predictions made by the model. From these values, key performance metrics, including accuracy, precision, recall, and F1 score, were computed. These metrics were calculated separately for the training and the testing datasets to assess the model’s performance during both the learning phase and on new, unseen data. Regular monitoring of these metrics throughout the model development process helped identify and address potential overfitting issues and ensured that the model maintained a good balance between precision and recall.

Ultimately, the importance of the variables that contributed to the prediction of the model classes was computed using Shapley values for each gene to estimate feature importance. The Shapley values are additive scale measures that represent the attribution of variables to the prediction. The Shapley values were computed using the SHAP package.

## 5. Conclusions

Overall, our inquiry enabled us to map the spatio-temporal gene expression of the different E1, E2, and HECT- and RBR-E3 UPS components in the human brain by analyzing a publicly available dataset. The outcomes revealed that the E1, E2, and HECT- and RBR-E3 family members’ gene expression patterns undergo dramatic changes at birth. Cluster analysis provided evidence highlighting that when human beings are exposed to external sensory stimuli (e.g., during infancy and early childhood) the E1, E2, and RBR- and HECT-E3 gene signature further changes. In addition, potential E1 → E2 → E3 cascades/axes were identified. Eventually, for the first time, by applying machine learning-based models to a brain-specific developmental transcriptome dataset, including fetal specimens, we validated and strengthened the findings, showing potential future applications of the analysis pipeline we employed for peculiar datasets characterized by data paucity and difficulty in retrieving biological material.

## Figures and Tables

**Figure 1 ijms-25-02361-f001:**
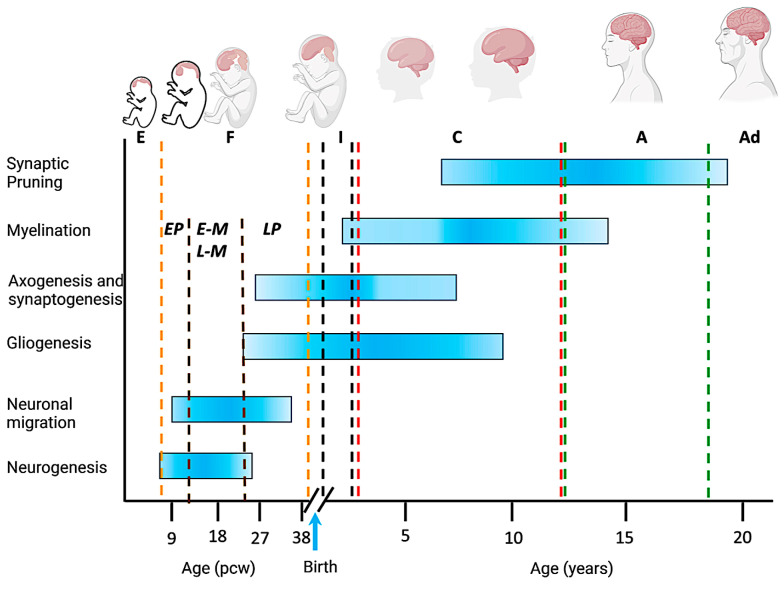
A timeline of human brain development during the prenatal (in post-conception weeks, pcw) and postnatal (in years) periods. The shaded horizontal bars represent the approximate timing of key neurobiological processes and developmental milestones. The vertical dashed lines define the different age groups. E: embryonic stage; F: fetal development; I: infancy; C: childhood (early childhood and late childhood); A: adolescence; Ad: adulthood; EP: early prenatal; E-M and L-M: early-mid and late-mid prenatal; LP: late prenatal. The blue vertical arrow indicates birth. Gross anatomical features and the relative size of the brain at different stages are illustrated at the top (Figure created with the help of BioRender, www.biorender.com, accessed on 1 November 2023).

**Figure 2 ijms-25-02361-f002:**
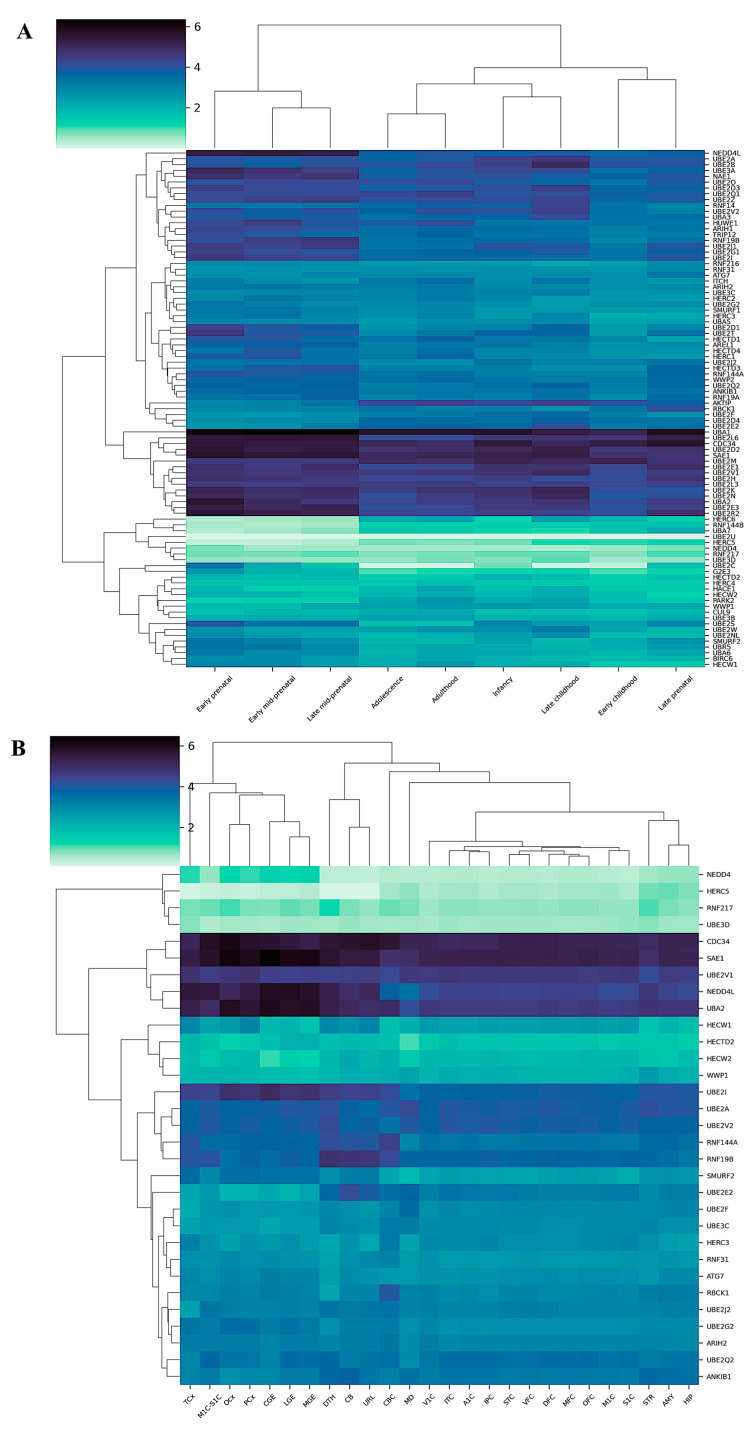
Hierarchical differential gene expression analysis. (**A**) Heatmap illustrating RNA-Seq differential expression data among different age groups and (**B**) among different brain regions. The heatmaps depict only those genes for which at least one condition displayed a significant difference when compared to all other conditions. AMY: amygdala; A1C: primary auditory cortex; CB: cerebellum; CBC: cerebral cortex; CGE: caudal ganglionic eminence; DTH: dorsal thalamus; DFC: dorsolateral prefrontal cortex; HIP: hippocampus; IPC: posteroventral (inferior) parietal cortex; ITC: inferolateral temporal cortex; LGE: lateral ganglionic eminence; M1C: primary motor cortex; M1C-S1C: primary motor–sensory cortex; MD: mediodorsal nucleus of thalamus; MFC: anterior cingulate medial prefrontal cortex; MGE: medial ganglionic eminence; OCX: occipital cortex; OFC: orbital frontal cortex; PCX: parietal neocortex; S1C: primary somatosensory cortex; STC: posterior superior temporal cortex; STR: striatum; TCX: temporal neocortex; V1C: primary visual cortex; VFC: ventrolateral prefrontal cortex; URL: upper (rostral) rhombic lip.

**Figure 3 ijms-25-02361-f003:**
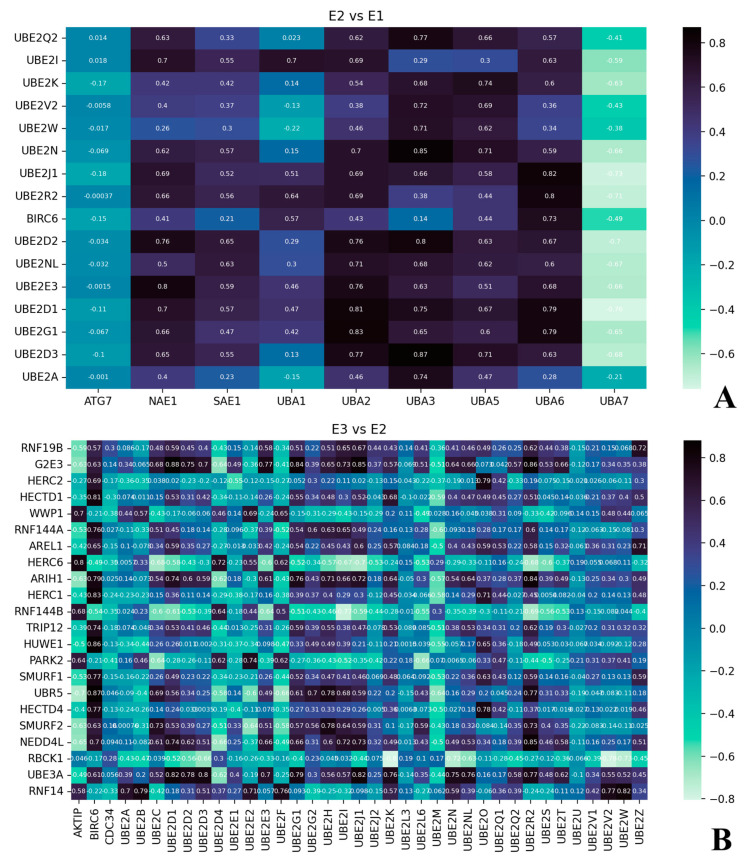
Correlation matrices. (**A**) Correlation matrix between E1 (plotted on the horizontal axis) and E2 (on the vertical axis) members and (**B**) matrix depicting the correlation between E2 (horizontal axis) and E3 (vertical axis) members. An arbitrary threshold of 0.7 was applied.

**Figure 4 ijms-25-02361-f004:**
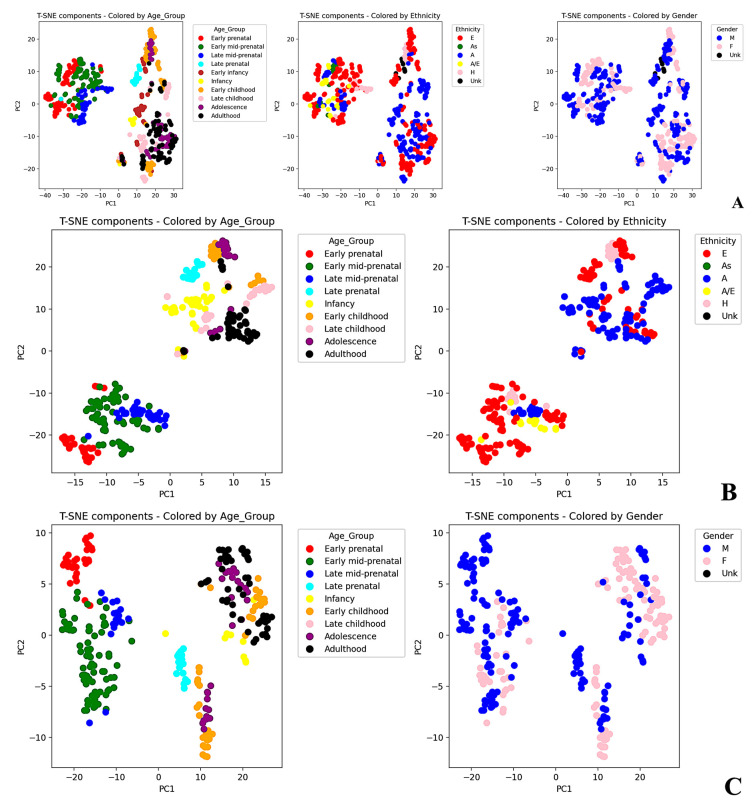
Dimensionality reduction with t-SNE. Dimensionality reduction of the data using t-SNE was applied in 2Ds to group of genes with similar expression patterns. Color codes were assigned to the clusters based on the categorical variable of interest, such as age, gender, and ethnicity. (**A**) The entire dataset was color-coded using the variables “age_group”, “ethnicity”, and “gender”, identifying two major sharp clusters corresponding to the prenatal and postnatal groups; (**B**) clustering analysis was restricted to a subset comprising only the “male” gender. In this case, observations were differentially color-coded by using the “age_group” and “ethnicity” variables; (**C**) the cluster survey was applied to a subset consisting exclusively of “European” ethnicity. In this latter case, the observations were color-coded with the “age_group” and “gender” variables. A: African; As: Asian; A/E: African/European; E: European; H: Hispanic; Unk: unknown.

**Figure 5 ijms-25-02361-f005:**
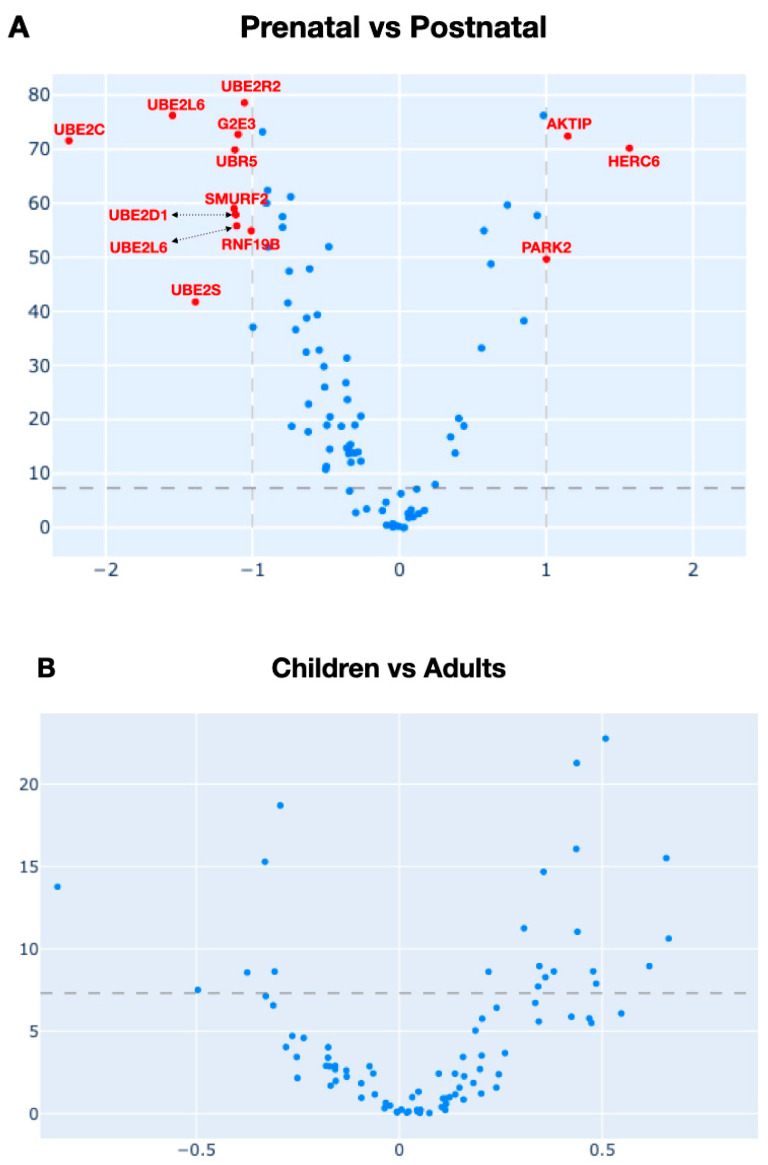
Differential gene expression between prenatal and postnatal groups and within the postnatal group. Volcano plot of RNA-Seq expression data for the prenatal and postnatal groups. The horizontal dashed line represents the significance threshold specified in the analysis, derived using a multiple testing correction (FDR adjustments through the Benjamin–Hochberg test). The dashed vertical lines bound the minimal fold change for the most differentially expressed genes. X-axis: log2 fold change; Y-axis: −log10-adjusted *p*-value. While the postnatal group, when compared to the prenatal group (**A**), exhibited a sharp and significant down-modulation of several E2 (e.g., *UBE2C*, *UBE2D1*, *UBE2L6*, etc.) and few E3 (e.g., *RNF19B*, *SMURF2*, *UBR5*, etc.), the subset analysis of the postnatal group (children vs. adult) did not show significant changes (**B**).

**Figure 6 ijms-25-02361-f006:**
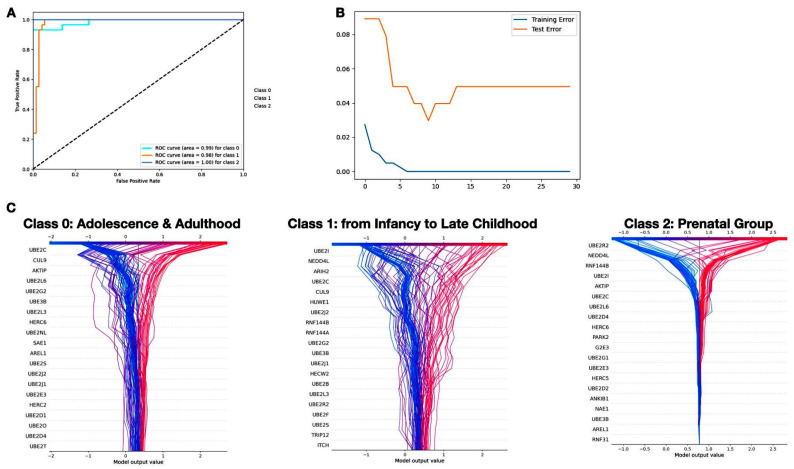
(**A**) ROC curve for XGBoost classification. The X-axis portrays the false positive rate (0 to 1), representing the proportion of actual negatives misclassified as positives. The Y-axis illustrates the true positive rate, indicating the proportion of actual positives correctly predicted as positives. Notably, the area under the curve (AUC) quantifies the overall performance, with AUC values of 0.99 (Class 0), 0.98 (Class 1), and a perfect 1.00 (Class 2); (**B**) evaluation of XGBoost performance over iterations. A compelling observation emerges, as both training and test datasets exhibit a noticeable plateau after 20 iterations. The graph illustrates a strikingly similar trend of the two, indicating stability in model performance. This plateau underscores the convergence of the XGBoost algorithm, suggesting optimal learning with diminishing returns beyond 20 iterations. (**C**) SHAP values feature importance analysis highlighting key genetic contributors with superior importance scores and shedding light on the influential role of top-ranking genes in the model’s predictive capacity.

**Figure 7 ijms-25-02361-f007:**
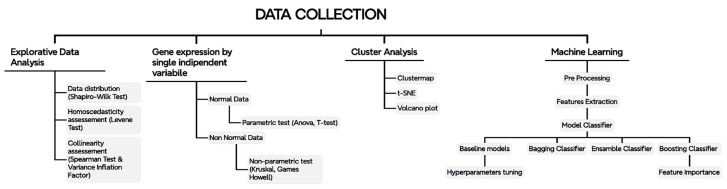
Integrated workflow pipeline—from raw data to machine learning classifications. The diagram flows from top to bottom and from left to right. The data collection, including data cleaning, was followed by an initial exploratory data analysis step with various checks of fundamental assumptions through the tests indicated in the figure. Then, data analysis for a single independent variable was performed. This implied clustering and visualization aspects (i.e., clustermap and volcano plot). Finally, machine learning was employed for target classification, as suggested by the clustering. The entire process is openly available on GitHub for complete transparency.

**Table 1 ijms-25-02361-t001:** Performance metrics—confusion matrix for the training and test sets using different classification models. The different models used are color-coded (KNN, pale yellow; SVC, white; RF, light blue; Voting Clf, pale green; XGBoost, pink), while the training and test sets’ rows are colored in white and light gray, respectively.

Model	Set	Precision	Recall	F1-Score	Accuracy	Support	Class
KNN	Train	0.91	091	0.91	0.95	114	0
		0.91	0.91	0.91		115	1
		1	1	1		172	2
	Test	0.9	0.97	0.93	0.96	29	0
		0.96	0.9	0.93		29	1
		1	1	1		43	2
SVC	Train	0.86	0.89	0.88	0.93	114	0
		0.89	0.86	0.88		115	1
		1	1	1		172	2
	Test	0.84	0.9	0.87	0.92	29	0
		0.89	0.83	0.86		29	1
		1	1	1		43	2
RF	Train	0.86	0.86	0.86	0.92	114	0
		0.85	0.85	0.85		115	1
		0.99	0.99	0.99		172	2
	Test	0.79	0.79	0.79	0.88	29	0
		0.79	0.79	0.79		29	1
		1	1	1		43	2
Voting Clf (SVC and KNN)	Train	0.9	1	0.95	0.97	114	0
		1	0.9	0.94		115	1
		1	1	1		172	2
	Test	0.93	0.97	0.95	0.97	29	0
		0.96	0.93	0.95		29	1
		1	1	1		43	2
XGBoost	Train	0.96	0.95	0.96	0.98	114	0
		0.95	0.97	0.96		115	1
		1	1	1		172	2
	Test	0.96	0.9	0.93	0.96	29	0
		0.9	0.97	0.93		29	1
		1	1	1		43	2

## Data Availability

For transparency the entire codebase and comprehensive workflow have been made openly accessible on GitHub (https://github.com/SMagnati/SMagnati, accessed on 1 March 2023). The data analysed in the study are from https://brainspan.org, accessed on 1 March 2023.

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
