# Peer review of "A Survey on the Expression of the Ubiquitin Proteasome System Components HECT- and RBR-E3 Ubiquitin Ligases and E2 Ubiquitin-Conjugating and E1 Ubiquitin-Activating Enzymes during Human Brain Development"

_ijms, 2024, doi:10.3390/ijms25042361_

Round 1

Reviewer 1 Report

Comments and Suggestions for Authors

The research report by Stefano Magnati et al. focuses on the expression of UPS components in human brain development. In this report, Correlation analysis is used to explain E1, E2, and E3 Ubiquitin and/or Ubiquitin Ligases. We report changes in the expression of several specific gene groups. This is thought to be able to greatly contribute to the elucidation of past brain development and future cranial nerve diseases.

There seems to be no problem with the method or the purpose of the paper. On the other hand, I would like to see improvements in the creation of figures and tables so that they can be more clearly understood by readers and contribute to improved understanding.

For example, black lines around each figure are not needed. Also, please make the figures larger so that the text is legible. For tables, please revise the title appropriately above the table and the explanation below.

Reviewer 2 Report

Comments and Suggestions for Authors

Please see the attached file for review comments.

Round 2

Reviewer 1 Report

Comments and Suggestions for Authors

The authors have made appropriate modifications and this is an improvement over previous versions. I have no other opinion.

Author Response

The authors express their warm thanks to the reviewer.

Sincereley

The authors

Stefano Magnati

Eugenio Alladio

Enrico Bracco

Reviewer 2 Report

Comments and Suggestions for Authors

Dear Authors,

Thank you for your reply to the review comments. Unfortunately, instead of corrected versions, old versions of both the manuscript and supplementary files have been attached. The "ijms-2815568-peer-review-v2" file is the same as the "ijms-2815568-peer-review-v1" file. The same applies to supplementary material files. For these reasons, until the files of both the manuscript and the supplementary materials are corrected, my comments remain unchanged.

Please address these important issues to enable the access to an improved manuscript and supplementary materials.

Sincerely,

Author Response

The authors regretted for the unpleasant inconvenience. We are bewildered for what has happened. For your own perusal, please find below enclosed the rebuttal.

We truly thank the reviewer for the insightful and constructive comments. We are glad that the manuscript has been appreciated overall. The issues, advice, and suggestions raised by the reviewer have been attentively and meticulously considered. The manuscript has now been revised according to your suggestions and we bring to your attention the revised form. We hope that the amendments produced have substantially improved the manuscript and thus to be reconsidered suitable for publication in the International Journal for Molecular Science.

In the revised version, all the modifications to the manuscript have been marked using the MS Word tracking function (red labeled).

Please find below enclosed our rebuttal with a point-by-point reply to the reviewer's comments and suggestions.

Annotations to the page numbers refer to the revised version of the manuscript.

R: refers to the reviewer’s comments, whereas A: to the authors’ reply in italics

R: The topic of the article “A survey on the expression of the UPS components Hect-, RBR-E3 Ub-ligase, E2 Ub-conjugating, and E1 Ub-activating enzymes during the human brain development” is interesting, however some important concerns need to be addressed before the manuscript is ready for publication in IJMS.

A: We are glad that the manuscript has been appreciated and found interesting by the reviewer. Indeed, we believe that being the first manuscript dealing with brain development and gene expression coupled with a machine-learning approach could pave the way for the investigation of the very early stages when paucity of material is a critical issue. 

Rebuttal point-by-point

General comments:

R1. The Introduction section should be corrected. Some important relevant references should be included.

A1: We thank the reviewer for the insight. For this purpose, we have now edited the “Introduction” section by making some amendments and by implementing the references and to some extent the text. The references added refer to brain development (page 3, lines 12-15; references #8-13; #8. Kinney, H.C.; Karthigasan, J.; Borenshteyn, N.I.; Flax, J.D.; Kirschner, D.A. Myelination in the Developing Human Brain: Biochemical Correlates. Neurochem Res 1994, 19, 983–996, doi:10.1007/BF00968708.; #9. Williamson, J.M.; Lyons, D.A. Myelin Dynamics Throughout Life: An Ever-Changing Landscape? Front. Cell. Neurosci. 2018, 12, 424, doi:10.3389/fncel.2018.00424.; #10. Grotheer, M.; Rosenke, M.; Wu, H.; Kular, H.; Querdasi, F.R.; Natu, V.S.; Yeatman, J.D.; Grill-Spector, K. White Matter Myelination during Early Infancy Is Linked to Spatial Gradients and Myelin Content at Birth. Nat Commun 2022, 13, 997, doi:10.1038/s41467-022-28326-4.; #11.  Qi, C.; Luo, L.-D.; Feng, I.; Ma, S. Molecular Mechanisms of Synaptogenesis. Front. Synaptic Neurosci. 2022, 14, 939793, doi:10.3389/fnsyn.2022.939793.: #12. Llorca, A.; Deogracias, R. Origin, Development, and Synaptogenesis of Cortical Interneurons. Front. Neurosci. 2022, 16, 929469, doi:10.3389/fnins.2022.929469.; #13. Selemon, L.D. A Role for Synaptic Plasticity in the Adolescent Development of Executive Function. Transl Psychiatry 2013, 3, e238–e238, doi:10.1038/tp.2013.7.), ubiquitination and ubiquitin ligases (page 3, lines 32-39; references #16-18; #16. Kravtsova-Ivantsiv, Y.; Ciechanover, A. Non-Canonical Ubiquitin-Based Signals for Proteasomal Degradation. Journal of Cell Science 2012, 125, 539–548, doi:10.1242/jcs.093567.; #17.  Kravtsova‐Ivantsiv, Y.; Sommer, T.; Ciechanover, A. The Lysine48‐Based Polyubiquitin Chain Proteasomal Signal: Not a Single Child Anymore. Angew Chem Int Ed 2013, 52, 192–198, doi:10.1002/anie.201205656.; #18.  Zheng, N.; Shabek, N. Ubiquitin Ligases: Structure, Function, and Regulation. Annu. Rev. Biochem. 2017, 86, 129–157, doi:10.1146/annurev-biochem-060815-014922.), and eventually to the section treating the proteasome machinery (page 3 lines 49-55 and page 4, lines 1-3; references #24-30; # 24. French, M.E.; Koehler, C.F.; Hunter, T. Emerging Functions of Branched Ubiquitin Chains. Cell Discov 2021, 7, 6, doi:10.1038/s41421-020-00237-y.; #25. Akutsu, M.; Dikic, I.; Bremm, A. Ubiquitin Chain Diversity at a Glance. Journal of Cell Science 2016, jcs.183954, doi:10.1242/jcs.183954.; #26. Dikic, I.; Schulman, B.A. An Expanded Lexicon for the Ubiquitin Code. Nat Rev Mol Cell Biol 2023, 24, 273–287, doi:10.1038/s41580-022-00543-1.; #27. Samant, R.S.; Livingston, C.M.; Sontag, E.M.; Frydman, J. Distinct Proteostasis Circuits Cooperate in Nuclear and Cytoplasmic Protein Quality Control. Nature 2018, 563, 407–411, doi:10.1038/s41586-018-0678-x.; #28. Yau, R.G.; Doerner, K.; Castellanos, E.R.; Haakonsen, D.L.; Werner, A.; Wang, N.; Yang, X.W.; Martinez-Martin, N.; Matsumoto, M.L.; Dixit, V.M.; et al. Assembly and Function of Heterotypic Ubiquitin Chains in Cell-Cycle and Protein Quality Control. Cell 2017, 171, 918-933.e20, doi:10.1016/j.cell.2017.09.040.; #29. Marshall, R.S.; Vierstra, R.D. Dynamic Regulation of the 26S Proteasome: From Synthesis to Degradation. Front. Mol. Biosci. 2019, 6, 40, doi:10.3389/fmolb.2019.00040.; #30. Rousseau, A.; Bertolotti, A. Regulation of Proteasome Assembly and Activity in Health and Disease. Nat Rev Mol Cell Biol 2018, 19, 697–712, doi:10.1038/s41580-018-0040-z.).

R2. The presentation of results should be improved.

A2: We have now amended all the figures (see below A3).

R3. Figures 2-7 are difficult to interpret (in many cases even at maximum magnification) because of too small font size/resolution and poor color contrast. This important issue should be addressed.

Perhaps the vertical orientation of the panels of some figures (2 and 3) would allow them to be enlarged.

A3: We deeply thank the reviewer for the insightful remark and suggestion. In the revised version of the manuscript, all the Figures have been edited. The figures have been magnified and for Figures 2, 3, and 5 the orientation has been switched from horizontal to vertical as advised. Figures 4, 6, and 7 to different extents have been magnified to improve their legibility. Table 1 has been modified by adding a color code, alongside the title and explanation were edited accordingly.

R4. Figures S1 A, S1 B, S1 C, S1 D, S1 E, and S1 I are not mentioned in the text of the manuscript (this important issue addressed).

A4: We are grateful to the reviewer for the remark, and we are frankly sorry about the omission. In the revised version of the manuscript, the gap has been filled. Now all supplementary figures are mentioned in the main text (page 10 lines 4-5). In addition, the caption for Figure S1 has been appended to the manuscript (page 21, lines 5-9).

R5. There are no page numbers in the PDF version of the article sent for review.

A5: We truly apologize for the oversight. We have now amended the manuscript by inserting page numbers.

Specific comments:

R6. Page 1, Abstract: It should be “the ubiquitin proteasome system (UPS)” instead of “the Ubiquitin Proteasome System (UPS)”.

A6: Page 1, ubiquitin proteasome system (UPS) has now been changed, as suggested by the reviewer, from uppercase to lowercase. 

R7. Page 1, Abstract: It should be “tagging with ubiquitin” instead of “tagging with Ubiquitin”.

A7: Page 1, also in this case ubiquitin has been edited from uppercase to lowercase.

R8. Page 1, Abstract: It should be “The E3 ubiquitin ligases” instead of “The E3 Ubiquitin Ligases“.

A8: Page 1, we have now switched E3 ubiquitin ligases from uppercase to lowercase as advised by the reviewer.

R9. Page 1, Keywords: It should be “ubiquitin proteasome system (UPS)” instead of “Ubiquitin Proteasome System (UPS)”.

A9: Page 1: in the revised form we have now edited the Keywords Ubiquitin Proteasome System from uppercase to lowercase.

R10. Page 2, Figure 1: Please compare “C: Childhood” marked in the figure with the age groups (“early childhood” and “late childhood”) described in subsection 4.1. and 4.3. of the Materials and Methods section (please see also Figure 4 for comparison). This issue should be addressed.

A10: We thank the reviewer for the remark, and we apologize for the small oversight. Indeed, by indicating Childhood we meant gathering the early and late childhood groups. We have now edited the oversight (page 2) by detailing the issue in brackets as follows “C: Childhood (early childhood and late childhood)”

R11. Page 2, the third line of the first paragraph: It should be “the evolutionary conserved ubiquitin proteasome system (UPS)” instead of “the evolutionary conserved Ubiquitin Proteasome System (UPS)”.

A11: Page 3, we have now amended the problem by switching from upper to lowercase.

R12. Page 2 (the last paragraph) and page 3 (the first paragraph): The role of 26S proteasomes in degradation of ubiquitinated proteins is not mentioned at all. This issue should be addressed. Relevant references should be included.

A12: We are very grateful to the reviewer for pointing out the deficiency. We apologize for it. In the revised version we have filled the gap by implementing the main text with the following (page 3) “Ubiquitination is a very versatile post-translational modification. Indeed, Ub can be conjugated directly to a target protein or to itself through either one of its seven conserved Lys residues (Lys6, Lys11, Lys27, Lys29, Lys33, Lys48 and Lys63) or N-terminal Met1 residue, thus leading to structurally distinct types of poly-Ub chains, linear or branched, of different length giving rise to an extremely broad Ub lexicon [24–26]. Usually, proteins modified with poly-Ub chains internally linked through K11 or K48 appear to be the favored proteasome substrates [27,28]. The 26S proteasome is a 2.5-MDa complex composed of different protein subunits arranged into an elongated structure composed of a central 20S core particle (CP) with one or two terminal 19S regulatory particle(s) (RP) [29,30]. Ubiquitinated proteins are recognized by the RP, and through an ATP-dependent mechanism are then unfolded, and then channeled to the catalytic CP, where the inner beta subunits, thanks to their proteolytic activities, ultimately degrade the target protein.”. Appended to the text there are also 7 new references.

R13. Page 3, the end of the first paragraph: I suggest “druggable candidates” instead of “druggable candidate”.

A13: Page the main text has been edited accordingly (page 3)

R14. Page 3, the end of the second paragraph: I suggest to mention more neurodegenerative disorders, besides Parkinson’s and cite a relevant article of the Nobel Prize laureate Aaron Ciechanover (Ciechanover, A., & Kwon, Y. T. (2015). Degradation of misfolded proteins in neurodegenerative diseases: therapeutic targets and strategies. Experimental & molecular medicine, 47(3), e147-e147).

A14: We thank the reviewer very much for the insightful advice. In the revised version the reference Ciechanover & Kwon has been appended (ref. 42, main text page 3).

R15. Pages 3-4, the subsection 2.1. of the Result section: Before discussing the results in detail, I recommend listing the relevant genes encoding enzymes of the ubiquitination machinery cascade (all the E1, E2, and E3 members) separately for each component.

A15: We acknowledge the reviewer for the suggestion. Consistently, in the revised version of the manuscript we provide a table, as Supplementary material (Table S1), in which the whole list of the genes surveyed is detailed and those discussed in subsection 2.1 are highlighted in yellow. The main text has been amended accordingly by adding the following sentence “The whole list of the genes surveyed and those displaying most noticeable changes is summarized in Table S1.” (page 4).

R16. Page 4: Italic fonts should be used for all genes names.

A16: we apologize for the blunder. In the revised version we edited gene names in italics as advised.

R17. Page 4, the Figure 2 caption: It should be “A)” and “B)” instead of “(A)” and “(B)” (please see figures 3, 4, and 6 for comparison).

A17: Thank you very much for noticing that and we are sorry for the inadvertence. As suggested by the reviewer now double brackets have been replaced by single brackets (page 6) for consistency with the other captions.

R18. Page 4, the Figure 2 caption: I suggest alphabetic ordering of the abbreviations used. Moreover, the abbreviations should be defined in the uniform manner (upper/lower case).19. Page 4, the Figure 2 caption: It should be “Dorsolateral Prefrontal Cortex” instead of “Dorsolateral Prefrontal Complex”.

A18: Thank you for the suggestion. In the current version in Figure 2 legend, the different brain regions have been ordered alphabetically, abbreviations edited uniformly and the typo concerning “Dorsolateral Prefrontal Complex” amended to “Dorsolateral Prefrontal Cortex”.

R20. Page 4 (the last line) and page 5 (the first line): Please verify the abbreviations “TCx, PCx, OCx” with those given in the Figure 2 caption.

A20: For consistency, in the revised version, the text has been edited and uniformed with the Figure 2 caption by swapping the lowercase “x” to uppercase (page 6).

R21. Page 5, subsection 2.2.: It should be “the E3 ubiquitin ligases” instead of “the E3 Ubiquitin Ligases“.

A21: Page 7, we edited from upper to lowercase the word ubiquitin, as suggested.

R22. Page 5, subsection 2.2.: I suggest “early childhood” instead of “early-childhood”.

A22: Page 7, the hyphen has now been removed.

R23. Page 6, the tenth line from the top: A necessary space is missing. It should be “Figure S1” instead of “FigureS1”.

A23: The text now has been slightly modified as follows “Remarkably, the survey outcome indicated robust correlations among different age groups, particularly in between the early childhood and the adolescence groups (Figure S1 F and H), while when the other age groups were analysed any significant correlations (Figure S1 A-E, G and I) was noticeable. “ (page 8). Hence, the typo has been edited.

R24. Page 7, the second and third lines from the bottom: The sentence “On the contrary, all the upregulated genes encode primarily for E3 members.” is a repetition of the previous sentence and should be deleted.

A24: We are sorry for the misprint. The repetition has now been deleted (page 10).

R25. Page 8, Figure 5: A symbol “A” is missing. There is no description for the left panel (“Prenatal vs Postnatal”) in Figure 5. Moreover, the dashed vertical lines are difficult to see because of poor color contrast.

A25: thank you very much for the notice. We apologize for the omission. The Figure has been implemented accordingly, by adding the “A”, by magnifying both panels, and thus vertical lines result in more visible.

R26. Page 8, Table 1: I suggest to modify the content of the title of Table 1. I suggest the first sentence (“Performance Metrics - Confusion Matrix for Train and Test Sets.”) as the table title. I propose to move the next three sentences (starting from “This table offers…”) to the paragraph above as a continuation of the sentence " Each model’s performance metrics concerning the training and testing sets are summarized in Table 1.".

A26: Thank you very much for the suggestion. The revised version has been changed as advised by moving the three sentences from the table’s caption to the Results (page 11) and by implementing the Table’s title as follows “Performance Metrics - Confusion Matrix for Train and Test Sets using different classification models: the different models used have been color-coded (KNN, pale yellow; SVC, white; RF, light blue; Voting Clf, pale green; XGBoost, Pink) while train and test sets rows are white and light gray colored, respectively” (page 11) because the table has been slightly changed to improve its clarity in terms of legibility and comprehensibility.

R27. Page 9, the fourth line from the bottom: I suggest “SHAP Llbrary” instead of “Shap Library”.

A27: As per suggestion Shap has been now written in uppercase (page 13).

R28. Page 10, the first and second line from the top: Necessary spaces are missing. It should be “(class 0)”, “(class 2)”, and “(Figure 6 C)” instead of “(class0)”, “(class2)”, and “(FIGURE6C), respectively.

A28: We thank the reviewer for pointing out these typos that we missed. We have now spaced the different classes (page 13).

R29. Page 10, the fourth line from the bottom: I suggest “genes encoding for the UPS members” instead of “gene encoding for the UPS members”.

A29: We thank the reviewer for the kind suggestion. Consistently, in the revised version the text has been edited (page 14).

R30. Page 10, the first line from the bottom: It should be “UBA1” instead of “UBA1”. Please see also comment no. 15.

A30: We thank the reviewer for the notice. We double-checked and edited.

R31. Page 11: Please verify the use of italics in the spelling of genes and proteins/enzymes.

A31: Thank you very much for the remark. We have now verified and edited; accordingly, genes in italics and proteins in regular font.

R32. Page 12, the third paragraph of subsection 4.1.: The number of age categories is incorrect. It should be “we grouped ages into nine representative categories: early prenatal, early mid-prenatal, late mid prenatal, late prenatal, infancy, early childhood, late childhood, adolescence, and adulthood.” instead of “we grouped ages into eight representative categories: early prenatal, early mid-prenatal, late-mid prenatal, late prenatal, infancy, early childhood, late childhood, adolescence, and adulthood.”.

A32: We are truly sorry for the blunder. In the revised version of the manuscript, we edited from eight to nine, as the reviewer pointed out (page 15).

R33. Page 12: An incorrect subsection number. It should be “4.3. Data analysis” instead of “4.2. Data analysis”.

A33: We are sorry for the mislabeling. In the revised version we have edited the numbering and in this specific case 4.3. replace the wrong 4.2. labeling.

R34. Page 12: Figure 7 is difficult to interpret because of too small font size (please see also comments no. 3 and 4).

A34: We thank the reviewer for bringing to our attention the issue. We have now slightly magnified Figure 7 depicting the pipeline used and attempted to improve the resolution.

R35. Pages 12-13: The spelling of subsection titles in the Materials and Methods section should be unified (please see the Results section for comparison).

A35: We are grateful again to the reviewer for spotting the typo that has now been edited accordingly (page 15).

R36. Page 13: An incorrect subsection number. It should be “4.4. Data validation” instead of “4.3. Data validation”.

A36: We are sorry for mislabeling again. In the revised version we have edited the numbering and in this specific case 4.4. replace the wrong 4.3. labeling.

R37. The Supplementary Materials section should be corrected. Any supplementary material published online alongside the manuscript should be described. Please indicate the name and title of each figure (please see the instruction for authors of IJMS for details).

A37: We apologize for the oversight. In the revised version of the manuscript, a legend to Figure S1 has been appended (page 17).

R38. The Author Contributions section: It should be “E.B.; data” instead of "E.B....; data"

A38: we truly apologize for this latest blunder. Punctuation has now been edited (page 17).

Furthermore, during the revision processes a few typos were detected and edited accordingly.

Round 3

Reviewer 2 Report

Comments and Suggestions for Authors

The revised version of the manuscript has been significantly improved. However, some concerns need to be addressed before the manuscript is ready for publication in IJMS. Please see the attached file for review comments.

Kind regards,

Author Response

We thank the Reviewer again for the detailed and accurate observations, and for giving us the possibility to further improve the clarity of our findings.

We considered all the comments of the Reviewers and changed the manuscript accordingly.

Please find attached the revised version of the original article entitled “A survey on the expression of the UPS components HECT-, RBR-E3 Ub-ligase, E2 Ub-conjugating, and E1 Ub-activating enzymes during the human brain development” which we are pleased to resubmit after careful revision.

The changes are embedded in the main text and have been marked using the MS Word tracking function (red labeled), whereas highlighted in yellow are the previous amendments.

Please find attached, as .pdf file, a detailed rebuttal to the issues raised.
